# CoMNet: Where Biology Meets ConvNets

## Abstract

Designing ConvNet and exploring its design space is a highly challenging research area. In this paper, inspired by the structural organization of cortical modules in the biological visual cortex, we present a pragmatically designed ConvNet architecture, called CoMNet which is simplified yet powerful. The bio-inspired design of CoMNet offers efficiency in multiple dimensions such as network depth, parameters, FLOPs, latency, branching, and memory budget at once while having a simple design space, in contrast to the existing designs which are limited only to fewer dimensions. We also develop a Multi-Dimensional Efficiency (MDE) evaluation protocol to compare models across dimensions. Our comprehensive evaluations show that in the MDE setting, CoMNet outperforms many representative ConvNet designs such as ResNet, ResNeXt, RegNet, RepVGG, and ParNet (Figure 1).

**Code:** Will be released post reviews.
**Supplemental:** See attachment.

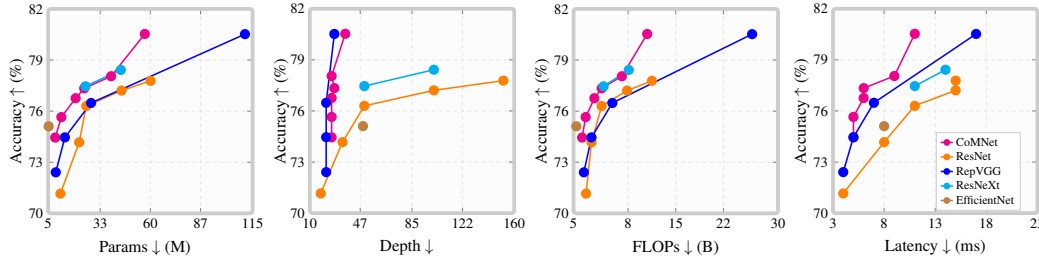

Figure 1: Multi-Dimensional Efficiency Results. For a model to be multi-dimensional efficient, it should always lie in the top-left region of the plot which is the case with the proposed CoMNet.

## 1 Introduction

Convolutional neural networks (ConvNets) remain important in terms of deployment, thanks to their hardware-friendly operations. However, ConvNet design (He et al., 2016; Liu et al., 2022) and its design space exploration (Radosavovic et al., 2020) remain a challenging problem. Existing approaches mainly focus on individual dimensions such as accuracy, FLOPs, and the number of parameters; however, addressing multiple dimensions *at once* is the current need.

This paper revisits ConvNets and presents a bio-inspired ConvNet design, namely CoMNet that surprisingly outperforms many representative ConvNets in multiple dimensions, even with a random choice of its design hyperparameters. CoMNet is our translation of biological underpinnings of cortical modules (Mountcastle, 1997), columnar organization (Mountcastle, 1997), pyramidal neurons and long-range connections (Mountcastle, 1997), predominantly found in the ventral stream of the biological visual cortex (Tanaka, 1996) that performs object recognition in mammals. These properties are fundamental to the cortex design and thus inspire our approach.

CoMNet offers lower architectural complexity, hardware-accelerator compatibility, low memory consumption, low memory access costs on parallel computing hardware, smaller depth, negligible branching, lower latency, low parameter and FLOPs *at once*. To the best of our knowledge, such a simplified design space while achieving Multi-Dimensional Efficiency (MDE) is rarely explored because it is a difficult task due to a high correlation among dimensions.

Although it is evident that the visual cortex inspired the earlier ConvNet designs (LeCun et al., 1998; Krizhevsky et al., 2012), many of its interesting properties are either missing or partially used in ConvNets. Most importantly, some of the cortex properties, such as weight sharing (Krizhevsky et al., 2012) or shortcut connections (He et al., 2016), have been explored individually. We comprehensively club valuable cortex properties into one architecture through a systematic study.

Summarily, our key contributions are:

1. A notion of Artificial Cortical Modules (ACM) which helps to achieve high representation in fewer parameters, controlled parameter growth, increased computational density (Sec. 4.1).
2. Studying columnar organization for smaller depths and lower latency (Sec. 4.2).
3. Studying long-Range Connections (LRC) similar to pyramidal neurons (Sec. 4.4).
4. Fusing the above principles into a single ConvNet design.
5. A notion of Multi-Dimensional Efficiency (MDE) protocol (Sec. 4.6).
6. Suggesting a design space of CoMNet. It is generally reported as a separate research work in ConvNets due to exhaustive effort demands (Sec. 4.8).

In the next section, we comprehensively discuss the most relevant works, followed by our biological insights (Sec. 3), and their translation into the CoMNet (Sec. 4). Then, in Sec. 5, we present a rigorous experimental analysis, and finally, in Sec. 6, we provide conclusions about the paper.

## 2 RELATED WORK

**Parameters and Representation Power.** The earlier CNNs (Krizhevsky et al., 2012; Simonyan & Zisserman, 2014; Szegedy et al., 2015; He et al., 2016) possess high representation power and use a large number of channels (Simonyan & Zisserman, 2014) in the deeper layers to compensate for the reduction in resolution, leading to exponential growth in parameters or synaptic connections of a kernel. It causes overfitting (Simonyan & Zisserman, 2014) that is handled via dropout but at the cost of more training epochs. ResNet (He et al., 2016) avoids that by channel squeezing and expanding via $1 \times 1$ convolutions. (Xie et al., 2017) partitions the ResNet blocks in the form of groups, however, the issue of large-depth, parameters are still intact.

MobileNets (Howard et al., 2017; Sandler et al., 2018; Zhang et al., 2018; Ma et al., 2018), on the other hand, reduce parameters and FLOPs by using depthwise convolutions (Sifre & Mallat) (DWC) that spans only a single channel. However, DWC reduces the representation power quickly and is devoid of cross-channel context which severely affects accuracy (Zhang et al., 2018). Therefore a DWC is followed by a $1 \times 1$ convolution to intertwine cross-channel context to improve accuracy.

**Depth.** The importance of networks being deeper is well analyzed (Liang & Srikant, 2017; Urban et al., 2017). However, the use of $1 \times 1$ convolutions exponentially increases the network depth, e.g., two $1 \times 1$ for each $3 \times 3$ in (He et al., 2016; Sandler et al., 2018; Zhang et al., 2018; Tan & Le, 2019) forming 66% of total depth, while one in (Howard et al., 2017) forming 50%. Moreover, their pointwise nature limits their contribution in the receptive field, which in contrast, is governed by $3 \times 3$ convolutions.

**Branching.** CNNs have grown from branchless (Krizhevsky et al., 2012; Simonyan & Zisserman, 2014) to single branch (He et al., 2016) to multi-branch (Szegedy et al., 2016; Radosavovic et al., 2020). Neural Architecture Search (NAS) has resulted in even heavily branched designs (Zoph et al., 2018; Tan & Le, 2019). Although branching improves accuracy (Srivastava et al., 2015), it significantly increases Memory Access Cost (MAC) on parallel computing hardware (Ding et al., 2021) which affects latency and memory consumption.

**Latency.** Both large depth and high branching increase latency even at fewer FLOPs and enough computing power because of the sequential layers and serialized execution of parallel branches where the output of one layer can not be computed until the output of its preceding layers is available. This dramatically increases latency even with fewer FLOPs per layer, e.g., 100 layers each of 1ms runtime result in 100ms latency while 15 layers each of 3ms runtime have 45ms latency. This phenomenon is prevalent in (Tan & Le, 2019) which has fewer FLOPs but runs equivalent to a five times bigger network (He et al., 2016).

Recent RepVGG (Ding et al., 2021) proposes structural reparameterization for accelerated inference, however, its train-time network has large parameters, branches, and training time, even more than its predecessor (He et al., 2016) (Table 2). More recently, ParNet (Goyal et al., 2021) built a shallower network to achieve lower latency, however, has exponentially high parameters even for an accuracy range of 77%, and also has branches within branches which, despite having fewer depth, are bound to be executed sequentially without any specialized implementations.

**Training Epochs:** Limited representation power (Howard et al., 2017; Sandler et al., 2018; Zhang et al., 2018; Ma et al., 2018; Tan & Le, 2019) needs longer train time and limits performance on downstream tasks, e.g., (Howard et al., 2017) requires 200 epochs on ImageNet and performs poorly on object detection in contrast to (Simonyan & Zisserman, 2014) which is trained only for 75 epochs. Similarly, (Tan & Le, 2019) requires 400 epochs and large resolution in contrast to (He et al., 2016; Simonyan & Zisserman, 2014; Radosavovic et al., 2020), which uses $90 - 120$ epochs range and smaller resolution $224 \times 224$. Hence, training duration is also a dimension in the MDE setting.

## 3 BIOLOGICAL VISUAL CORTEX

A biological visual cortex is a fairly complex structure with several interesting properties. We highlight the most relevant ones below. For more details, please refer to the supplement.

**Columnar Structure.** Cortical modules are present all across the visual cortex (Mountcastle, 1997). The cortical modules in the shallower layers are referred to as ocular dominance columns that respond to simple stimuli such as edges and lines of different orientations (Hubel & Wiesel, 1963). In contrast, modules in the deeper layers are a collection of neurons that respond to complex stimuli, such as the face, monkey, human, etc., by encoding stimuli information from different viewpoints (Tanaka, 1996). These modules are mainly present in the Inferotemporal cortex (IT) that is responsible for object detection and recognition tasks (our inspiration).

**Shared Input or Input Replication.** Multiple cortical modules having different stimuli responses can have a common input, i.e., the input is shared or replicated. This is intuitive since a given location in the visual field may contain any stimuli, i.e., a monkey or a car. Hence, multiple modules work in parallel, and the one having the highest similarity with the stimuli fires strongly and signals to other parts of the cortex (Hubel & Wiesel, 1963).

**Limited Synaptic Connections.** A cortical column contains only a few neurons ($70 - 100$) (Mountcastle, 1997), resulting in fewer synaptic connections. This collection of neurons can learn simple stimuli.

**Massive Parallelization.** Regardless of layers being shallower or deeper, a cortical module processes only a small region of the retina. Multiples of such modules with similar stimuli responses are replicated to span the retinal field. This organization facilitates massive parallelization.

**Lateral Connection Inhibition.** Cortical modules can not communicate with each other, except at their output (Tanaka, 1996) via pyramidal neurons that have a large number of long-range connections and fuse cross-module information (Tanaka, 1996) to gather larger spatial context and to learn better representations.

Some of these ideas, e.g., massive parallelization, have been used in ConvNets in the form of weight sharing or convolution, but not all of them are explored altogether in a single ConvNet design. In this paper, we club the above ideas into a single ConvNet design, our major novelty. Our intention in this paper is not to compete with much more carefully designed architectures via trial-and-error with complicated training schemes such as (Liu et al., 2022) or Transformers (Dosovitskiy et al., 2020; Liu et al., 2023), which all together work on different structures. Instead, we aim to develop an improved template ConvNet design, which outperforms prominently used ConvNets in the community.

## 4 COMNET

To build CoMNet, we first combine the above fundamental design attributes of the cortex design and visualize them in Figure 2a. Then we translate these attributes or Figure 2a into its neural equivalent, as shown in Figure 2b. From there, we develop its CNN equivalent (Figure 2c), and finally, we develop the fundamental computational unit of CoMNet, called CoMNet-unit (Figure 2d).

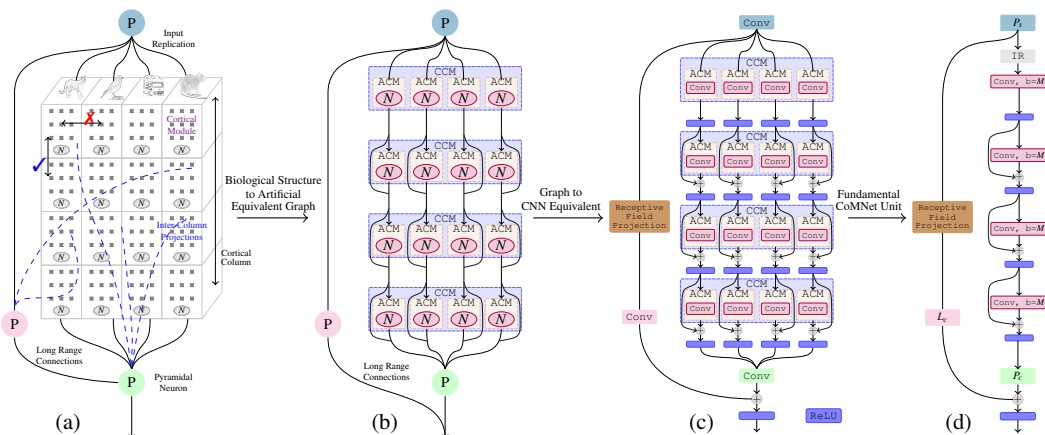

Figure 2: Our translation of biological underpinnings into CoMNet. (a) Cortical module structure in a cortex Tanaka (1996), (b) from biological to artificial-equivalent graph, (c) from graph to the CNN-equivalent, and (d) CoMNet-unit. *P*: Pyramidal neurons. *N*: number of neurons in a cortical column. IR: Input replication.

## 4.1 CORTICAL MODULES

To realize IT-like structure in CNNs, we develop an Input-Replication mechanism (IR) and Artificial Cortical Modules (ACM). Figure 2b shows our translation of a bio-cortical module into an ACM.

IR transforms a tensor $\in \mathbb{R}^{C \times H \times W}$ into duplicated one $\in \mathbb{R}^{(M \times C) \times H \times W}$, where $M$ denotes the desired number of cortical modules. This operator returns $M$ identical replicas of the input (Figure 3).

Since a cortical module is essentially a group of neurons, we realize its CNN equivalent via a $k \times k$ convolution (conv) having $N$ neurons, where $k \in \mathbb{R}_{\geq 3}$ (Figure 2c). By following the limited synaptic connection property, $N$ is kept small. For $M$ cortical modules, $M$ convs are employed in parallel, each processing one of the replicas obtained from the IR. We refer $M$ ACMs as a Collective Cortical Module (CCM) where parallelly operating ACMs indicate its cardinality.

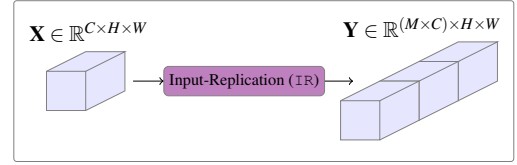

Figure 3: Illustration of Input-Replication (IR).

## 4.2 COLUMNAR STRUCTURE

In a cortical column, neurons are connected both parallelly and serially (Mountcastle, 1997). The parallel connections get realized implicitly via $N$ neurons in a module. However, to realize the serial ones, we stack ACMs to form a column (Figure 2b). Then, to respect lateral connection inhibition, communication between ACMs is allowed only within a column (Figure 2a). Since the CNN equivalent of ACM is a convolution, multiple convolution layers are stacked to realize the columnar behavior. These convolutions do not communicate with the convolutions in other columns to respect lateral connection inhibition.

## 4.3 PYRAMIDAL NEURONS

Pyramidal neuron (Mountcastle, 1997) is a crucial entity in the visual cortex having a large number of synapses. It serves different purposes, such as fusing the output of multiple columns, feeding subsequent columns, or facilitating long-range projections (Mountcastle, 1997). Due to its importance, we also translate this idea to CNNs.

We realize a pyramidal neuron via $1 \times 1$ convolution and use it for different purposes, similar to the bio-pyramidal neurons; for input summarization that feeds the IR denoted as $P_s$, for fusing inter-column information denoted as $P_c$, and for long-range connections denoted as $L_c$ (discussed next). $P_s$ is fed by the output of previous network stages and feeds IR, whereas $P_c$ is fed by the

output of multiple cortical columns or simply the final CCM, and fuses the input by operating at each $(h, w) \in \mathbb{R}^{(M \times N) \times H \times W}$. Each neuron in the $P_c$ has many connections, arising from combining the output of $M \times N$ channels, which greatly mimics a pyramidal neuron.

### 4.4 LONG RANGE CONNECTIONS

Pyramidal neurons also project their input to many layers (long range) (Mountcastle, 1997) that helps exploit multi-layer information. To realize this behavior, we use a $1 \times 1$ convolution $L_c$ that is fed by the output of the preceding CoMNet-unit (discussed next). $L_c$ projects its input to the output of the unit, where it is fused with the output of $P_c$. This simulates the behavior of combining the cortical module information and multi-layer information (Figure 2b). Since $L_c$ directly connects the input and output of a CoMNet while bypassing all the columns, it mimics Long-Range Connections (LRC).

In CNNs, parameter growth is a crucial issue if left uncontrolled, and long-range bio-pyramidal neurons have a very large number of connections to obtain a large receptive field (Mountcastle, 1997). Directly adapting this behavior may result in a large number of connections. To avoid that, we devise a Receptive Field Projector (RFP), and instead of feeding $L_c$ directly, we first feed RFP, and then from RFP to $L_c$. RFP essentially is $k \times k$ pooling where $k \in \mathbb{R}_{\geq 2}$. The pooling operation increases the receptive field of $L_c$ by summarizing the neighborhood in the input. Otherwise, due to the point-wise nature of $L_c$, it becomes difficult to offer a large receptive field to $L_c$. Although LRC improves accuracy significantly, they can be traded for parameters and FLOPs at the cost of reduced accuracy.

### 4.5 COMNET-UNIT

The above-proposed design translations finally lead to the fundamental computational unit of CoMNet, called CoMNet-unit (Figure 2d). Such a unit is fed by a tensor $T_i$, passed through $P_s$, producing a tensor $T_s$ having channels reduced by a factor $\zeta$. $T_s$ is then passed through IR followed by a stack of CCMs, connected via residual connections (He et al., 2016) to prevent vanishing gradients.

Whereas $P_c$ is fed the output of final CCM. $T_i$ is also fed to $L_c$ preceded by RFP, denotes as $p = $ RFP$(T_i)$), which projects the previous stage i.e. $T_i$ to the $P_c$. Now, the output of $P_c$ and $L_c$ are summed to produce an output tensor $T_o$. Mathematically, the whole CoMNet-unit can be written as follows:

$$T_s = P_s(T_i), \quad T_{\text{IR}} = \text{IR}(T_s), \quad T_{ccm} = \text{CCM}(.), \quad T_c = \text{CCM}_l \odot ...\text{CCM}_1 \odot \text{CCM}_0(T_{\text{IR}}) \tag{1}$$

$$T_o = P_c(T_c) + L_c(p(T_i)) \tag{2}$$

### 4.6 MULTI-DIMENSIONAL EFFICIENCY (MDE)

Considering practical utility and deployment of a model for a given accuracy range, we focus on five most crucial dimensions: *latency*, *depth*, *branching*, *FLOPs*, and *parameters*, because they are sufficient to achieve multi-dimensional efficiency of controlled parameter growth, high representation power in fewer parameters, high computational density, minimal branching, while other objectives such as memory consumption, memory access cost depend on these.

**Controlled Parameter Growth.** In the earlier CNNs, the number of parameters is changed by altering network width or depth, which offers less precise control and leads to exponential parameter growth, especially in the deeper layers (Sec. 2). CoMNet has several ACMs which have only a few neurons and fewer synaptic connections due to their operation being confined to only one column. Hence, altering the width or depth of CoMNet affects its number of parameters less aggressively (Table A1 in Appendix). This flexibility is quite crucial during scaling a CNN as per the requirements.

**High Representation Power.** We hypothesize that multiple neurons with fewer connections are better than a single neuron with large connections. Any stimuli requires a certain amount of representation power or neurons to be learned. In the existing CNNs, a kernel operates on a large number of channels and thus has a large number of synaptic connections. During the learning phase, it gets penalized for all the visual stimuli even if it is uncorrelated with them, also verifiable via weight pruning (Li et al., 2016) that eliminating many channels/connections does not impact accuracy until a certain point, indicating a wastage of synaptic connections.

CoMNet counters this issue by synthesizing more neurons out of a single large neuron, which helps CoMNet achieve higher accuracy in fewer parameters without prolonging the training time (Table A2 in Appendix) in contrast to (Sandler et al., 2018; Tan & Le, 2019), indicating better representations learned by CoMNet. (Figure A1 in Appendix)

**Increased Parallelization and Computational Density.** Since all of the ACMs in a CCM are operating parallelly, they can be processed efficiently by using NVIDIA's CUDA-based highly optimized `Batched-Matrix-Multiply` routines. To achieve that, we combine all ACMs of a CCM into a single convolution having $M$ batches. This strategy packs computations of all ACMs into a single convolution, which leads to increased computational density, increased GPU utilization, and reduced memory access cost (Ding et al., 2021). Thus resulting in a much simplified CoMNet design (Figure 2d). A Pytorch code snippet that implements the CoMNet unit is shown in Sec. F Appendix.

**Reduced Depth, FLOPs, and Latency.** As mentioned in Sec. 2, $1 \times 1$ layers are the major constituent of depth in state-of-art networks because they are based on blocks which are a stack of $1 \times 1$, $3 \times 3$, and $1 \times 1$ layers. Several such blocks are connected serially to form a stage. For instance, a stage with three units has six $1 \times 1$ layers for three $3 \times 3$ layers. Since the receptive field is mainly governed by $k \times k$ convolution layers with $k \in \mathbb{R}_{>1}$ (Luo et al., 2016), the columnar organization facilitates the elimination of $1 \times 1$ layers by stacking $3 \times 3$ CCM layers, achieving equivalent receptive in just three layers. In other words, the columnar organization reduces the three blocks into one.

The elimination of $1 \times 1$ convolutions drastically reduces network depth, resulting in reduced FLOPs and latency (Sec 2). For instance, CoMNet performs better than ResNet-50 at 50% fewer layers while having lower parameters, FLOPs, and latency, indicating a huge achievement.

**Minimal Branching and Memory Access Cost.** CoMNet is uni-branched regardless of training and testing. It is important since it reduces per-iteration training time, memory consumption, and memory access costs. This contrasts with recent RepVGG (Ding et al., 2021) that are inferior during training.

**Hardware Acceleration.** Since a CoMNet-unit is made up mostly of $3 \times 3$ convolutions, it well suits the CNN hardware accelerators because they have dedicated support for them.

**Faster Convergence.** In just half epochs i.e. 60, CoMNet achieves 99.17% (76.16%) of its accuracy obtained at 120 epochs (76.76%), as compared to ResNet-50 which achieves only 97% (74.15% vs 76.32%). We believe that it happens because $3 \times 3$ convolutions are more important since they solely govern the receptive field in contrast to $1 \times 1$. Hence, CoMNet obtains an equivalent receptive field by only using $3 \times 3$ convolutions. However, $3 \times 3$ convolutions have an overly large number of parameters for which $1 \times 1$ layers were employed that increased the network depth(He et al., 2016). Our CoMNet tackles this issue implicitly via ACMs.

## 4.7 RELATION WITH EXISTING DESIGNS

Here, we discuss how some of the ideas we inherit from the cortex are also in use in existing ConvNets and how our instantiation of these ideas differs from them.

**Input replication (IR)** The idea of input replication is quite common in the cortex. In ConvNets, this was first proposed in Inception (Szegedy et al., 2015), then in ResNeXt (Xie et al., 2017). After that, this idea has not been in use in the designs popular in the community since, in these designs, it caused inefficiency. For example, Inception uses different-sized convolutions and pooling after replication. Therefore, every single unit needs to be executed serially, although they are employed in parallel. The idea of IR in ResNeXt is more similar to us however, the major difference is that ResNeXt has multiple blocks per stage, and each performs replication; on the other hand, CoMNet performs input replication only once and has much deeper columns. Similarly, Inception is not a columnar architecture since it does not have deeper columns.

**Group Convolutions.** Although group convolutions are widely explored (Xie et al., 2017; Zhang et al., 2018), there are two key differences. *First*, group convolution divides the input channels, thus defying the objective of input replication because now each column receives only a subset of the input channels thus less information per group. On the contrary, CoMNet uses IR, which feeds each column with the replica of the input, thus making the entire input information accessible to each column.

*Second*, group convolutions are followed by $1 \times 1$ layers to avoid loss of accuracy due to the lack of inter-group communication (Zhang et al., 2018). This increases network depth and, hence, latency. On the contrary, CoMNet is free from such constraint, which fuses the columns only once via $P_c$. We analyzed what would happen if CoMNet also uses the same strategy at the same parameter/FLOP budget by keeping constant accuracy. It comes out that it increases the network depth and latency (Sec. C).

**Long Range Connections.** The long-range connections find structural similarity with projections in (He et al., 2016). However, there are notable differences: *First*, projections in (He et al., 2016) are used only in the first block of a stage, and projection between stages does not exist. *Second*, projection operates at a stride of 2. In CoMNet, $L_c$ is preceded by a receptive field projection (RFP) which gathers spatial context for $L_c$, and CoMNet aligns the spatial size of the input with $P_c$.

**No Blocks Only Stage.** Interestingly, our CoMNet does not have blocks, unlike modern CNNs, which have stages, and each stage comprises multiple blocks (He et al., 2016; Xie et al., 2017; Goyal et al., 2021; Liu et al., 2022). As an example, ResNet-50 has four residual stages, having 3, 4, 6, and 3 blocks respectively. On the contrary, CoMNet has only the notion of a stage, i.e., a CoMNet-unit is essentially a stage of CoMNet which turns CoMNet design significantly simpler and offers many benefits as discussed previously. Overall, CoMNet structure is very different from the existing CNNs (See Figure A4 in Appendix). Based on the above discussion, we claim that although the inherited ideas, especially the input replication and columnar organization, may be seen as existing in the previous ConvNets. However, our use of these ideas is entirely different from theirs, resulting in a new structure of CoMNet. Moreover, it is also visible that none of the previous works uses all of the ideas together in a way CoMNet does, differentiating CoMNet design from the existing ones.

### 4.8 COMNET INSTANTIATION

A CoMNet variant can be instantiated by sequentially connecting CoMNet-units. Without complicating, we follow earlier designs (He et al., 2016; Simonyan & Zisserman, 2014) to keep the tradition of five stages, among which first is a plain $3 \times 3$ convolution with a stride of 2, while remaining are the CoMNet-units. Following (He et al., 2016), we set channels of $P_s$ to 64, which gets doubled at each stage, while the channels of $P_c$ and $L_c$ always equal to $\zeta$ times that of $P_s$. We set $\zeta = 4$, following (He et al., 2016).

To further simplify the instantiation, we set the number of CCM layers, i.e. $l$ in $k^{th}$ CoMNet-unit equal to the number of blocks in the $k^{th}$ stage of RESNET-50 (He et al., 2016), a widely used model. A CoMNet-unit has *only three hyperparameters: $M, N, l$*. In this work, we do not explore the whole space since it requires massive computing resources and months of duration (Radosavovic et al., 2020). Even the earlier CNNs (He et al., 2016; Simonyan & Zisserman, 2014) and the newer ones (Ding et al., 2021) avoid exploring the whole design space.

For this reason, given CoMNet is easy to configure, we could train only fewer models that put CoMNet into the context of most representative state-of-the-art models (See Table A1 in Appendix) and can outperform them in MDE setting.

## 5 EXPERIMENTS

### 5.1 IMAGENET CLASSIFICATION

**Training Protocol.** We test CoMNet on ImageNet (Deng et al., 2009) benchmark. We train CoMNet variant for 120 epochs using SGD, Nesterov momentum, base_lr=0.1 with cosine-scheduler (Loshchilov & Hutter, 2016), and RandomResized crop (Paszke et al., 2019) and random flip.

**MDE Testing Protocol.** Based on the practical significance, we define a dimension precedence i.e. Latency = Depth > Branching > FLOPs > Parameters which helps fair evaluation when efficiency is not possible in all dimensions. Although parameter efficiency is essential, it can be sacrificed if a model is better in other dimensions while latency is kept at the highest priority.

**Main Results.** We show that CoMNet achieves multi-dimensional efficiency in a large spectrum of models while being simpler during both training and inference and offering competitive trade-offs

Table 1: CoMNet at standard 120 epochs schedule. Latency @ RTX-2070 GPU that may vary for other GPU, hence the numbers are only for reference.

| Row | Architecture | #Depth ↓ | #Params ↓ | FLOPs ↓ | Latency ↓ | FPS ↑ | Top-1 (%) ↑ |
|-----|-------------|----------|-----------|---------|-----------|-------|-------------|
| R0 | ● ResNet-18 He et al. (2016) | 18 | 11.6M | 1.83B | 4ms | 250 | 71.16 |
|     | ● ResNet-34 He et al. (2016) | 34 | 21.7M | 3.68B | 8ms | 125 | 74.17 |
|     | ● **CoMNet-A0** | 26 | **8.8M** | **1.25B** | 7ms | 142 | **74.45** |
| R1 | ● EfficientNet-B0 Ding et al. (2021) | 49 | 5.26M | 0.40B | 8ms | 125 | 75.11 |
|     | ● **CoMNet-A1** | **26** | 12.1M | 1.77B | **7ms** | **142** | **75.65** |
| R2 | ● ResNet-50 Paszke et al. (2019) | 50 | 25.5M | 4.12B | 11ms | 90 | 76.30 |
|     | ● **CoMNet-B1** | **26** | **19.8M** | **3.05B** | **7ms** | **143** | **76.76** |
| R3 | ● ResNet-101 He et al. (2016) | 101 | 44.5M | 7.85B | 15ms | 67 | 77.21 |
|     | ● ResNeXt-50 Xie et al. (2017) | 50 | 25.1M | 4.4B | 11ms | 90 | 77.46 |
|     | ● **CoMNet-C1** | **28** | **24.4M** | **4.12B** | **7ms** | **143** | **77.34** |
| R4 | ● ResNet-152 He et al. (2016) | 152 | 60.1M | 11.5B | 15ms | 67 | 77.78 |
|     | ● ResNeXt-101 Xie et al. (2017) | 101 | 44.1M | 8.10B | 14ms | 71 | 78.42 |
|     | ● ParNet-L Goyal et al. (2021) | 12 | 55M | 26.7B | 23ms | 43 | 77.66 |
|     | ● ParNet-XL Goyal et al. (2021) | 12 | 85M | 41.5B | 25ms | 40 | 78.55 |
|     | ● **CoMNet-C2** | **26** | **38.9M** | **7.09B** | **11ms** | **90** | 78.05 |

relative to the rival network. We compare CoMNet in two settings; at standard training (120 epochs) with CNNs (Table 1), and with Structural Reparameterization (Ding et al., 2021) separately (Table 2).

**Comparison with standard CNNs.** As shown in Table 1 R0, CoMNet is 3.29% more accurate, has 25% fewer parameters, shows similar runtime, and shows 31% fewer FLOPs than RESNET-18 although CoMNet has 6 more layers. Similarly, in contrast to RESNET-34, it is more accurate by 0.28% with 59% fewer parameters, 66% fewer FLOPs, and 23% less layers, while it is fast by 37%. RESNET-50 is the widely employed backbone such as (Ren et al., 2015; He et al., 2017; Carion et al., 2020; Goyal et al., 2017) due to its affordability in terms of representation power, FLOPs, depth, and accuracy. Table 1 R2 shows that CoMNet easily surpasses RESNET-50 while being 50% shallower, 22% fewer parameters, 25% fewer FLOPs, and 40% faster.

Although we do not aim mobile regime in this paper, we show that having fewer parameters and FLOPs does not guarantee faster speeds. As shown in Table 1 R1, EFFICIENTNET-B0 has 50% fewer parameters and 77% fewer FLOPs, but is 50% deeper, and runs 37% slower. By exploring the design space of CoMNet, we hope CoMNet can be extended to the mobile regime.

As shown in Table 1 R3-R4, CoMNet is better than bigger variants of RESNET, which still serves as backbones for cutting-edge works (Carion et al., 2020; Li et al., 2022). Our CoMNet outperforms them in every aspect while being 72% and 82% less deep relative to RESNET-101 and RESNET-152, respectively. CoMNet also runs faster by 50% in 50% fewer parameters and FLOPs. In addition, despite being smaller than RESNEXT (Xie et al., 2017), CoMNet outperforms it in all the dimensions. Overall CoMNet is 50% less deeper than RESNEXT-50 while running 50% faster at 6% fewer FLOPs, 2% fewer parameters while being more accurate. In contrast to RESNEXT-101, CoMNet is 75% less deeper, 11% fewer parameters, 12% fewer FLOPs, and 35% faster at a higher accuracy.

Moreover, CoMNet even outperforms recent non-deep PARNET (Goyal et al., 2021) that only aims at lower latency. CoMNet is uni-branched, while PARNET has multiple shallow branches which serialize the computations, thus making them deeper virtually.

**Comparison with RepVGG.** REPVGG (Ding et al., 2021) uses *Structural Reparameterization* during inference to offer plain VGG-like (Simonyan & Zisserman, 2014) structure. However, its training complexity is very high due to a large number of parameters and three branches at each layer (Ding et al., 2021), which increases the training time (Table 2). Compared with REPVGG family, CoMNet offers considerably lower complexity during both training and testing, thanks to its CCM layers. In addition, CoMNet has fewer parameters and fewer FLOPs while offering similar speeds with higher accuracy. Note that REPVGG-B3 is dramatically less efficient than CoMNet, i.e., 116% more parameters, 143% more FLOPs while running slower.

**Faster Convergence.** Table 3 R0-R1 shows the faster convergence of CoMNet in half of the original training epochs, i.e., 60. It can be seen that CoMNet attains 99.17% (76.16%) of its accuracy obtained at 120 epochs (76.76%), as compared to RESNET-50 which achieves only 97% (74.15% vs 76.32%).

Table 2: CoMNet *vs* recent Structural Reparameterization (SR) of RepVGG (Ding et al., 2021). #SR_Params, #SR_FLOPs, and #SR_Latency are the metrics with structural reparameterization. CoMNet is simple in both training and testing.

| Row | Architecture | #Depth ↓ | #Epochs | #Params ↓ | #SR_Params ↓ | #FLOPs ↓ | #SR_FLOPs ↓ | Latency ↓ | SR_Latency ↓ | FPS ↑ | SR_FPS ↑ | Top-1 (%) ↑ |
|---|---|---|---|---|---|---|---|---|---|---|---|---|
| R0 | ● RepVGG-A0 | 22 | 120 | 9.1M | 8.30M | 1.51B | 1.46B | 8ms | 4ms | 125 | 250 | 72.41 |
| | ● **CoMNet-A0** | 26 | 120 | **8.8M** | 8.80M | **1.25B** | **1.25B** | **7ms** | 5ms | **143** | 200 | **74.45** |
| R1 | ● RepVGG-A1 | 22 | 120 | 14.0M | 12.7M | 2.63B | 2.36B | 7ms | 5ms | 143 | 200 | 74.46 |
| | ● RepVGG-B0 | 28 | 120 | 15.8M | 14.3M | 3.06B | 3.40B | 7ms | 5ms | 143 | 200 | 75.14 |
| | ● **CoMNet-A1** | 26 | 120 | **12.1M** | 12.1M | **1.77B** | **1.77B** | 7ms | 5ms | 143 | 200 | **75.65** |
| R2 | ● RepVGG-A2 | 22 | 120 | 28.1M | 25.5M | 5.69B | 5.12B | 9ms | 7ms | 111 | 143 | 76.48 |
| | ● **CoMNet-B1** | 26 | 120 | **19.8M** | 19.8M | **3.05B** | **3.05B** | **7ms** | **6ms** | **143** | 167 | **76.76** |
| | ● **CoMNet-C1** | 28 | 120 | **24.4M** | 24.4M | **4.12B** | **4.12B** | **7ms** | **6ms** | **143** | 167 | **77.34** |
| R3 | ● RepVGG-B3 | 28 | 200 | 123.0M | 110.9M | 29.1B | 26.2B | 22ms | 17ms | 45 | 58 | 80.52 |
| | ● **CoMNet-D1** | 36 | 200 | **57.0M** | 57.0M | **10.8B** | **10.8B** | **12ms** | **11ms** | **83** | 90 | **80.53** |

Table 3: CoMNet demonstration for faster convergence. CoMNet can quickly reach high accuracy only in a very few epochs in comparison to ResNet-like models He et al. (2016).

| Row | Architecture | #Depth ↓ | #Epochs | #Params ↓ | #FLOPs ↓ | Latency ↓ | Top-1 (%) ↑ | Δ-Top-1 ↓ |
|---|---|---|---|---|---|---|---|---|
| R0 | ● ResNet-50 | 50 | 60 | 25.5M | 4.12B | 10ms | 74.15 | – |
| | ● ResNet-50 | 50 | 120 | 25.5M | 4.12B | 10ms | 76.30 | 2.15% |
| R1 | ● **CoMNet-B1** | 26 | 60 | 19.8M | 3.05B | 7ms | 76.16 | – |
| | ● **CoMNet-B1** | 26 | 120 | 19.8M | 3.05B | 7ms | 76.76 | **0.60%** |

Table 4: CoMNet with attention mechanism i.e. SE Hu et al. (2018), CBAM Woo et al. (2018), AFF Dai et al. (2021), and SKNet Li et al. (2019).

| | Approach | #Epochs | #Depth ↓ | #Params ↓ | #FLOPs ↓ | Top-1 (%) ↑ |
|---|---|---|---|---|---|---|
| R0 | ● ResNet-50 + SE | 120 | 50 | 28.09M | 4.13B | 76.85 |
| | ● ResNet-50 + CBAM | 120 | 50 | 28.09M | 4.13B | 77.34 |
| | ● **CoMNet-B1** | 120 | 26 | **19.20M** | **3.05B** | 76.77 |
| | ● **CoMNet-B1** + SE | 120 | 26 | **20.10M** | 3.10B | **77.85** |
| R1 | ● ResNet-50 + AFF | 160 | 50 | 30.30M | 4.30B | 79.10 |
| | ● ResNet-50 + SKNet | 160 | 50 | 27.70M | 4.47B | 79.21 |
| | ● **CoMNet-C1** + SE | 160 | 28 | **25.01M** | **4.13B** | **79.51** |

**Conjunction with Attention Mechanisms.** Table 4 R0-R1 shows that when CoMNet is used in conjunction with Squeeze and Excitation (SE) like attention (Hu et al., 2018), it outperforms recent attention mechanism (AFF (Dai et al., 2021), SKNET (Li et al., 2019), and CBAM (Woo et al., 2018)) in the MDE setting.

**Advanced CNNs and Transformers.** We conduct experiments with modern networks. Please see Sec. G in Appendix for additional results.

## 6 CONCLUSION

We propose *CoMNet* which focuses on CNN design from the perspective of multi-dimensional efficiency. CoMNet inherits key properties of a biological visual cortex such as *cortical modules, columnar organization, pyramidal neurons* to achieve multi-dimensional efficiency in parameters, FLOPs, accuracy, latency, and training duration at once while having simple architecture. We provide a minimal design space of CoMNet instead of only a few models. CoMNet outperforms many representative CNNs such as ResNet, ResNeXt, RegNet, RepVGG, and ParNet, while being shallower, faster, and offering competitive trade-offs when multi-dimensional efficiency is not possible.

**Limitations.** Despite the achievements, CoMNet is open for improvement. In this paper, we have only built a simple template architecture that can be further evolved like (Liu et al., 2022). For instance, a comprehensive design space of CoMNet including mobile regime can be explored, similar to (Radosavovic et al., 2020), or BrainScore can serve as an additional efficiency dimension.

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

Table A1: Minimal design space of CoMNet with multi-dimensional performance.

| Model | $P_c$ | | | | $N$ | | | | $l$ | | | | $M$ | | | | #Depth | #Params | #FLOPs | Latency | #Epochs | Top-1 (%) |
|---|---|---|---|---|---|---|---|---|---|---|---|---|---|---|---|---|---|---|---|---|---|---|
| ● CoMNet-A0 | 256 | 512 | 1024 | 2048 | 16 | 32 | 64 | 128 | 3 | 4 | 6 | 3 | 1 | 1 | 1 | 1 | 26 | 8.8M | 1.25B | 4ms | 120 | 74.45 |
| ● CoMNet-A1 | 256 | 512 | 1024 | 2048 | 16 | 32 | 64 | 128 | 3 | 4 | 6 | 3 | 4 | 4 | 4 | 4 | 26 | 12.1M | 1.77B | 5ms | 120 | 75.65 |
| ● CoMNet-A2 | 256 | 512 | 1024 | 2048 | 16 | 32 | 64 | 128 | 3 | 4 | 6 | 3 | 5 | 5 | 5 | 5 | 26 | 13.2M | 1.95B | 7ms | 120 | 76.01 |
| ● CoMNet-B0 | 256 | 512 | 1024 | 2048 | 32 | 64 | 128 | 256 | 3 | 4 | 6 | 3 | 1 | 1 | 1 | 1 | 26 | 11.3M | 1.65B | 4ms | 120 | 75.37 |
| ● CoMNet-B1 | 256 | 512 | 1024 | 2048 | 32 | 64 | 128 | 256 | 3 | 4 | 6 | 3 | 4 | 4 | 4 | 4 | 26 | 19.8M | 3.05B | 6ms | 120 | 76.76 |
| ● CoMNet-B2 | 256 | 512 | 1024 | 2048 | 32 | 64 | 128 | 256 | 3 | 4 | 12 | 3 | 4 | 4 | 4 | 4 | 32 | 23.3M | 3.74B | 8ms | 120 | 77.15 |
| ● CoMNet-B3 | 256 | 512 | 1024 | 2048 | 32 | 64 | 128 | 256 | 3 | 4 | 6 | 3 | 5 | 5 | 5 | 5 | 26 | 22.6M | 3.51B | 6ms | 120 | 77.01 |
| ● CoMNet-C0 | 256 | 512 | 1024 | 2048 | 48 | 80 | 144 | 272 | 3 | 4 | 6 | 3 | 4 | 4 | 4 | 4 | 26 | 21.6M | 3.73B | 6ms | 120 | 76.93 |
| ● CoMNet-C1 | 256 | 512 | 1024 | 2048 | 48 | 80 | 144 | 272 | 4 | 4 | 6 | 4 | 4 | 4 | 4 | 4 | 28 | 24.4M | 4.12B | 7ms | 120 | 77.34 |
| ● CoMNet-C1 | 256 | 512 | 1024 | 2048 | 48 | 80 | 144 | 272 | 4 | 4 | 6 | 4 | 4 | 4 | 4 | 4 | 28 | 24.4M | 4.12B | 7ms | 200 | 78.54 |
| ● CoMNet-C2 | 256 | 512 | 1024 | 2048 | 48 | 80 | 144 | 272 | 3 | 4 | 6 | 3 | 6 | 6 | 16 | 6 | 26 | 38.9M | 7.09B | 9ms | 120 | 78.05 |
| ● CoMNet-C2 | 256 | 512 | 1024 | 2048 | 48 | 80 | 144 | 272 | 3 | 4 | 6 | 3 | 6 | 6 | 16 | 6 | 26 | 38.9M | 7.09B | 9ms | 200 | 79.25 |
| ● CoMNet-D0 | 256 | 512 | 1024 | 2048 | 64 | 96 | 160 | 288 | 3 | 4 | 6 | 3 | 4 | 4 | 4 | 4 | 26 | 23.6M | 4.52B | 8ms | 120 | 77.25 |
| ● CoMNet-D1 | 256 | 512 | 1024 | 2048 | 64 | 96 | 160 | 288 | 4 | 5 | 12 | 5 | 4 | 4 | 12 | 4 | 36 | 57.0M | 10.8B | 11ms | 200 | 80.53 |

# APPENDIX

## A  DESIGN SPACE

Table A1 shows a few CoMNet instances.

## B  ADDITIONAL EVALUATIONS

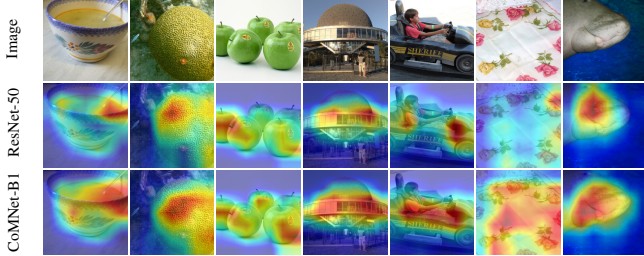

Figure A1: GradCAM visualizations

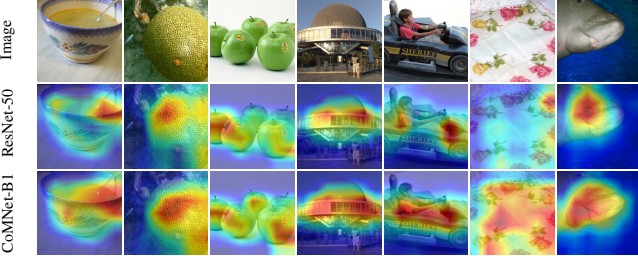

Figure A2: Axiom-GradCAM Fu et al. (2020) visualizations

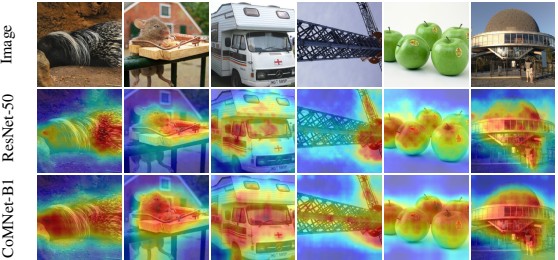

Figure A3: Full Gradient CAM Srinivas & Fleuret (2019) visualizations

**GradCAM, Axiom-CAM, and Full-Grad-CAM.** To comprehend why CoMNet performs better in MDE setting, we investigate its class activation maps on ImageNet (Deng et al., 2009) validation set

Table A2: Effect of ACM, Varying $M, N, l$, and LRC. Values of $M, N, l$ are for each of the four CoMNet stages.

| Row | N | | | | l | | | | M | | | | #Depth | LRC | Residual CCM | #Params | #FLOPs | Top-1 (%) |
|---|---|---|---|---|---|---|---|---|---|---|---|---|---|---|---|---|---|---|
| R0 | • 16 | 32 | 64 | 128 | 3 | 4 | 6 | 3 | 1 | 1 | 1 | 1 | 26 | ✓ | ✓ | 8.80M | 1.25B | 74.45 |
| R1 | • 16 | 32 | 64 | 128 | 3 | 4 | 6 | 3 | 4 | 4 | 4 | 4 | 26 | ✓ | ✓ | 12.1M | 1.77B | 75.65 |
| R2 | • 16 | 32 | 64 | 128 | 3 | 4 | 6 | 3 | 5 | 5 | 5 | 5 | 26 | ✓ | ✓ | 13.2M | 1.95B | 76.01 |
| R3 | • 32 | 64 | 128 | 256 | 3 | 4 | 6 | 3 | 1 | 1 | 1 | 1 | 26 | ✓ | ✓ | 11.3M | 1.65B | 75.37 |
| R4 | • 32 | 64 | 128 | 256 | 3 | 4 | 6 | 3 | 4 | 4 | 4 | 4 | 26 | ✓ | ✓ | 19.8M | 3.05B | 76.76 |
| R5 | • 32 | 64 | 128 | 256 | 3 | 4 | 6 | 3 | 5 | 5 | 5 | 5 | 26 | ✓ | ✓ | 22.6M | 3.51B | 77.01 |
| R6 | • 32 | 64 | 128 | 256 | 3 | 4 | 6 | 3 | 1 | 1 | 1 | 1 | 26 | ✓ | ✗ | 11.3M | 1.65B | 75.28 |
| R7 | • 32 | 64 | 128 | 256 | 3 | 4 | 6 | 3 | 1 | 1 | 1 | 1 | 26 | ✓ | ✓ | 11.3M | 1.65B | 75.37 |
| R8 | • 32 | 64 | 128 | 256 | 4 | 5 | 20 | 3 | 1 | 1 | 1 | 1 | 44 | ✓ | ✗ | 13.4M | 2.12B | 75.18 |
| R9 | • 32 | 64 | 128 | 256 | 4 | 5 | 20 | 3 | 1 | 1 | 1 | 1 | 44 | ✓ | ✓ | 13.4M | 2.12B | 75.88 |
| R10 | • 32 | 64 | 128 | 256 | 3 | 4 | 6 | 3 | 1 | 1 | 1 | 1 | 26 | ✗ | ✓ | 8.5M | 1.29B | 73.61 |
| R11 | • 32 | 64 | 128 | 256 | 3 | 4 | 6 | 3 | 1 | 1 | 1 | 1 | 26 | w/o. RFP | ✓ | 9.8M | 1.44B | 74.15 |
| R12 | • 32 | 64 | 128 | 256 | 3 | 4 | 6 | 3 | 1 | 1 | 1 | 1 | 26 | w. RFP | ✓ | 9.8M | 1.44B | 75.37 |

via GradCAM (Selvaraju et al., 2017), Axiom-CAM (Fu et al., 2020) and Full-Grad-CAM (Srinivas & Fleuret, 2019) which computes regions attended by the network for a given class. CAM visualizations of ResNet-50 and CoMNet-B1 (based on R2, Table 1) are shown in Figure A1, A2, A3. It can be seen that CoMNet is better at learning to attend regions of the target class relative to the baseline. It indicates the high generalizability of CoMNet by emphasizing class-specific parts in the input image.

**BrainScore.** As the structure of IT cortex is our major inspiration, we ran BrainScore (Schrimpf et al., 2020) on CoMNet-B1 out of curiosity. The score provides a degree to which the intermediate layers of a network behave similarly to the layers of the visual cortex (Schrimpf et al., 2020). In our test, we found that CoMNet ranks 39 (in more than 350 entries) in the IT score without any specialized training. The models ResNet-101 and ResNet-152 are ranked 119 and 143 respectively in the IT score.

## C   ABLATION STUDY

**Varying M and N.** Table A2 demonstrates the effect of varying $N$ and $M$ (R0-R5). We first fix the values of $N$ and vary $M$ (R0-R5), and then vary $M$ while fixing $N$ (R0 ↔ R3, R1 ↔ R4, R2 ↔ R5). It can be seen that for fixed $N$, accuracy improves by increasing $M$, and the same effect is seen by fixing $M$ while varying $N$. It can be noticed that parameters, FLOPs can be precisely controlled by changing the $M$ (R1 ↔ R2, R4 ↔ R5) which directly reflects accuracy.

**Effect of ACM.** We compare instances having different $N, M$, but have similar parameters and FLOPs budget, for instance, R1 ↔ R2, R1 ↔ R3, Table A2. It is noticeable that $R2$ with 5 ACM is better by 0.36% in accuracy, only at 1.1M more parameters relative to R1. Similarly, $R1$ is better by 0.28% in accuracy, only at 0.8M more parameters relative to R3. It shows that multiple ACMs facilitates improved accuracy in just a fraction of parameters and FLOPs. Moreover, if comparing R9 (a deeper model) with R2, R2 achieves 0.13% more accuracy in 0.2M fewer parameters and 0.17B fewer FLOPs. It shows the advantage of having multiple cortical modules while being shallower.

**Varying l.** The impact of varying $l$ is shown in R9, Table A2. It can be seen that going deeper is not necessary because a shallower version with same parameters (R2) is more accurate. Moreover increased depth causes increased latency in R9, therefore, we stick to $20 - 40$ layers of depth. We also carry out additional experiments where each CCM is followed by a $1 \times 1$ convolution as done in group convolutions while keeping depth and parameters constant. We observe 1% accuracy drop.

**Residuals in CCM.** R6-R9, Table A2 shows this analysis. For the shallower model, the residual connection shows only minor improvement (0.09%), however, for the deeper model, the effect of residual connections is noticeable (0.70%).

**Effect of Long Range Connections (LRC).** We train a CoMNet instance in three ways: *First*, remove LRC entirely, *Second*, use LRC without receptive field projection (RFP), and *Third*, LRC with RFP. See Table A2 for the analysis. It can be noticed that without LRC (R10), the model suffers with heavy accuracy loss of $\sim 0.54\%$ relative to when LRC is used without RFP (R11). Moreover, when using LRC with RFP (R12), accuracy improves, i.e., 1.22% and 1.76% relative to R11 and R10,

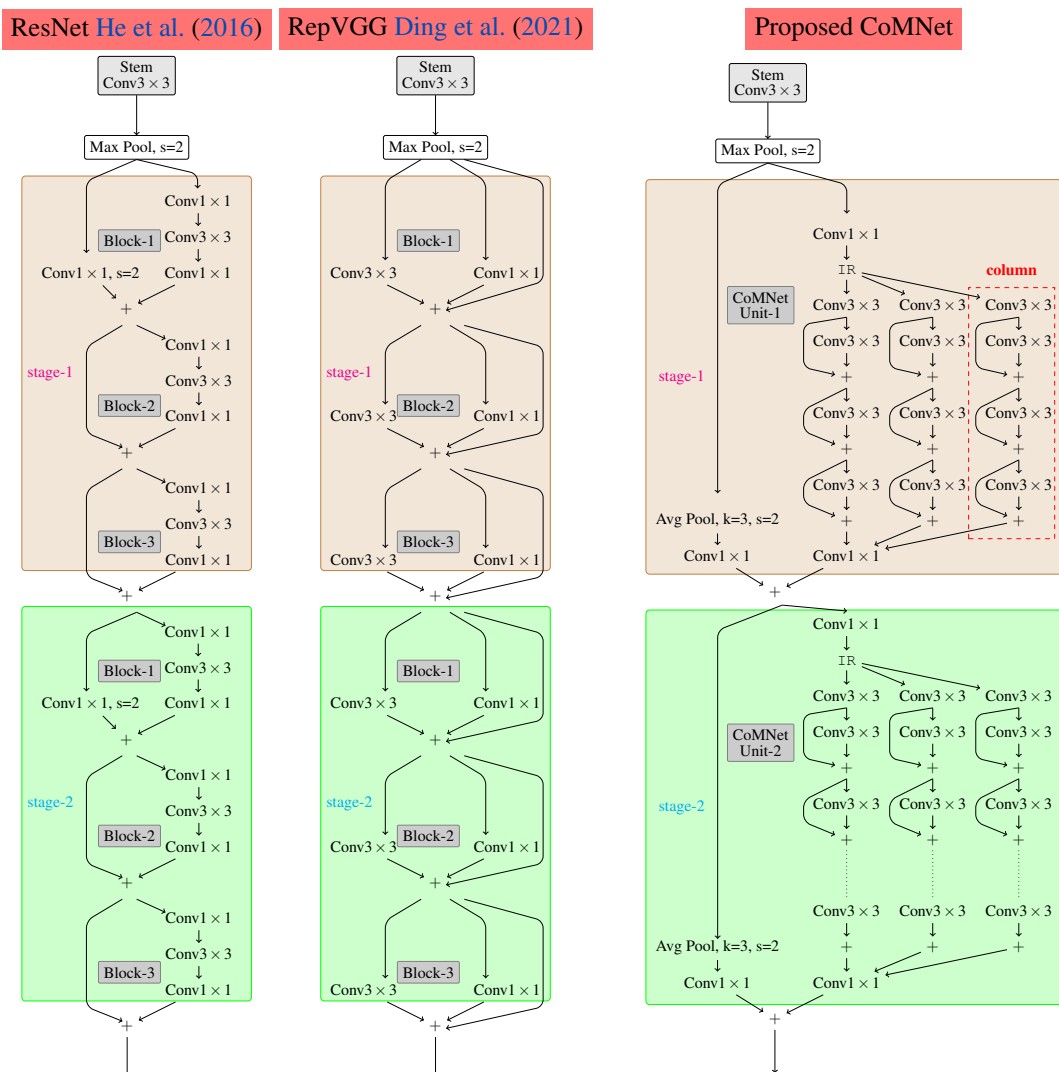

Figure A4: Illustrative figure of representative architectures in contrast to CoMNet. Notice that many convolutional neural network architectures share the same structure of ResNet He et al. (2016) such as ResNeXt Xie et al. (2017), RegNet Radosavovic et al. (2020) etc. Here, we have visualized three main architectures. Notice that A CoMNet-unit has one stage which does not have blocks, whereas, in other networks, multiple blocks are repeated. In this figure, there are many branches in CoMNet which are shown just to visualize and differentiate CoMNet from other networks at fundamental level. These branches in real implementation converges into one branch since we perform batched convolution. The exact number of $3 \times 3$ layers may differ depending on the configuration.

respectively, because `RFP` provides more spatial context to the $1 \times 1$ $L_c$ layer by summarizing the neighborhood.

## D  STRUCTURAL COMPARISON WITH EXISTING NETWORKS

Figure A4 shows architecture differences between CoMNet and the existing representative models (He et al., 2016) and (Ding et al., 2021). It can be seen that a CoMNet unit is a stage, whereas ResNet and RepVGG-like models have blocks and, thus, large depths. See Figure A4 for a better understanding.

Figure A5 shows architecture differences between CoMNet and inception models.

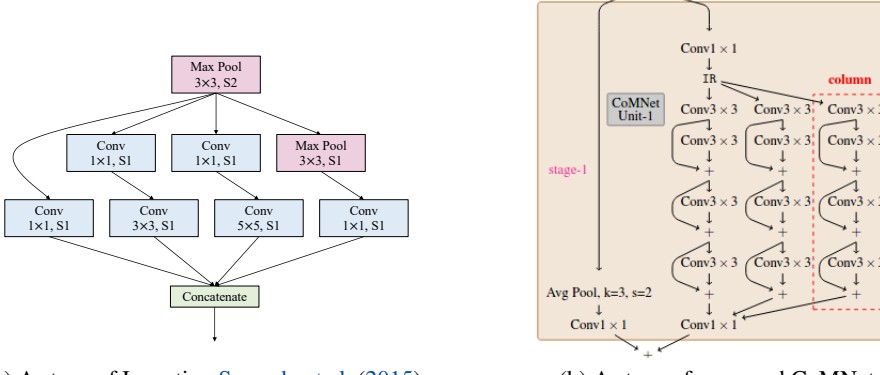

(a) A stage of Inception Szegedy et al. (2015).  (b) A stage of proposed CoMNet.

Figure A5: Difference between Inception Szegedy et al. (2015) and CoMNet stage.

# E  TRAINING SETTING

We train models in PyTorch (Paszke et al., 2019). We use eight NVIDIA RTX-2070 GPUs.

# F  PYTORCH CODE

All codes shall be open-sourced in PyTorch (Paszke et al., 2019) post the review process. Here, we provide a code snippet of a CoMNet Unit. Please see until the end of this document.

```python
class InputReplicator(nn.Module):
    def __init__(self, M):
        super(InputReplicator, self).__init__()

        # number of ACM
        self.M = M

    def forward(self, ip):
        x = ip.repeat(1, self.M, 1, 1)
        return x

class CoMNetUnit(nn.Module):
    def __init__(self, n_ip):
        super(CoMNetUnit, self).__init__()

        self.n_op_Pc = 256 # 512, 1024m 2048
        self.N = 32  # acm --> cortical module
        self.stride = 2
        self.l = 3 # 4, 6, 3]
        self.M = 4# 4, 4, 4]

        n_op_Ps = int(self.n_op_Pc / 4)

        self.conv_ps = nn.Conv2d(n_ip, n_op_Ps, 1, 1, 0, bias=False)
        self.bn_ps = nn.BatchNorm2d(n_op_Ps)
        self.relu_ps = nn.ReLU(True)

        self.IR = InputReplicator(self.M)

        # we limit the n_op of last CCM layer so that the parameters of the 1x1 expansion layer
        # do not grow overly large if number of modules is very big
        # as a rule of thumb, we set it nearly equal to  n_op / 4
        self.n_op_ccm_last = int(round(n_op_Ps / self.M)) * self.M

        self.conv_ccm = nn.ModuleList()
        self.bn_ccm = nn.ModuleList()
        self.relu_ccm = nn.ModuleList()

        self.conv_ccm.append(nn.Conv2d(n_op_Ps * self.M, self.N * self.M, 3, self.stride, 1,
                                       groups=self.M, bias=False))
        self.bn_ccm.append(nn.BatchNorm2d(self.N * self.M))
```

```
44              self.relu_ccm.append(nn.ReLU(True))
45
46          for i in range(self.l-2):
47              self.conv_ccm.append(nn.Conv2d(self.N * self.M, self.N * self.M, 3, 1, 1,
48                                      groups=self.M, bias=False))
49              self.bn_ccm.append(nn.BatchNorm2d(self.N * self.M))
50              self.relu_ccm.append(nn.ReLU(True))
51
52          self.conv_ccm.append(nn.Conv2d(self.N * self.M, self.self.n_op_ccm_last, 3, 1, 1,
53                                  groups=self.M, bias=False))
54          self.bn_ccm.append(nn.BatchNorm2d(self.n_op_ccm_last))
55          self.relu_ccm.append(nn.ReLU(True))
56
57          self.conv_pc = nn.Conv2d(self.n_op_ccm_last, self.n_op_Pc, 1, 1, 0, bias=False)
58          self.bn_pc = nn.BatchNorm2d(self.n_op_Pc)
59          self.relu_pc = nn.ReLU(True)
60
61
62          self.conv_lc = nn.Conv2d(n_ip, self.n_op_Pc, 1, 2, 0, bias=False)
63          self.bn_lc   = nn.BatchNorm2d(self.n_op_Pc)
64
65      def forward(self, ip):
66          x = self.relu_ps(self.bn_ps(self.conv_ps(ip)))
67          x = self.IR(x)
68
69          x = self.relu_ccm[0](self.bn_ccm[0](self.conv_ccm[0](x)))
70
71          for i in range(1, self.l - 2):
72              y = self.bn_ccm[i](self.conv_ccm[i](x))
73              x = self.relu_ccm[i](x + y)
74
75          # Last CCMN needs to handled with care because n_op for last CCMN may not match
76          # with n_op of the previous CCM layer
77          # and thus an idenity residual connection is not possible
78          # In other words, a residual connection will be used iff n_op of all CCM layers
79          # is same
80          if (self.N * self.M == self.n_op_ccm_last):
81              idx = self.l - 1
82              y = self.bn_ccm[idx](self.conv_ccm[idx](x))
83              x = self.relu_ccm[idx](x + y)
84          else:
85              idx = self.l - 1
86              x = self.relu_ccm[idx](self.bn_ccm[idx](self.conv_ccm[idx](x)))
87
88          x = self.bn_lc(self.conv_lc(x))
89
90          z = F.avg_pool2d(ip, 3, 2, 1)
91          z = self.bn_lc(self.conv_lc(z))
92
93          return F.relu_lc(x + z)
94
95
```

# G    ADDITIONAL RESULTS

Table A3: CoMNet comparison at longer training schedule i.e. $200 - 300$ epochs with larger models of Transformers and state-of-the-art CNN models. FPS is not throughput, instead it is *per-sample runtime*, metric of practical significance. $*$ denotes retrained baselines for fair comparison from official repositories.

| Row | Architecture | Type | #Depth ↓ | #Epochs ↓ | #Params ↓ | #FLOPs ↓ | Latency ↓ | FPS ↑ | Top-1 (%) ↑ |
|---|---|---|---|---|---|---|---|---|---|
| R0 | ● DieT-S Touvron et al. (2021) | Txfmr | 48 | 300 | 22M | 4.6B | 15ms | 66 | 79.80 |
|  | ● **CoMNet-B3** | CNN | **26** | 300 | 22M | **3.5B** | **6ms** | **167** | **79.95** |
|  | ● Swin-T- Liu et al. (2021)* | Txfmr | 96 | 300 | 28M | 4.5B | 20ms | 50 | 80.10 |
|  | ● ConvNeXt-T Liu et al. (2022)* | CNN | 59 | 300 | 29M | 4.5B | 13ms | 77 | 81.80 |
| R1 | ● RegNetX-12GF Radosavovic et al. (2020) | CNN | 57 | 200 | 46M | 12.1B | 13ms | 77 | 80.55 |
|  | ● **CoMNet-B4** | CNN | **26** | 300 | 30M | 5.1B | **9ms** | **111** | **81.25** |

**Comparison with advanced CNNs and Transformers.** As shown in Table A3 R0, CoMNet is almost 50% less deep, has 23% fewer params, and runs 60% faster than DIET Transformer while exhibiting slightly better accuracy. In Table A3 R1, CoMNet is 72% less deeper, 55% faster, and 1.15% more accurate than the popular SWIN Transformer while is 55% less deeper, 30% faster with slightly lower accuracy than the popular CONVNEXT models, which is an achievement in MDE setting.

