# OpenReview forum: "CoMNet: Where Biology Meets ConvNets"
_ICLR.cc/2024/Conference — Submitted to ICLR 2024_

### Official Review · Reviewer_mQDh · 2023-10-28

**Soundness:** 3 good
**Presentation:** 3 good
**Contribution:** 3 good
**Rating:** 5
**Confidence:** 4

**Summary:**

ComNet is a convolutional image classification network, with inspiration for its operations drawn heavily from biological structure of columnar organization.  In particular, ComNet uses stacked group convs to implement structures that parallel column neurons.  The result is a relatively simple network that performs well, more efficient (in multiple cost measures) than relevant baselines using similar structures for both training and inference.  Measurements were performed on ImageNet.

**Strengths:**

The system is well-motivated, and simple while achieving good performance.  The parallel with columns is an interesting way to view group convs.  I also like the analogies with biological neuron organization --- while a lot of papers allude to loose parallels or look at specific low-level behaviors, I think this paper makes concrete links for nearly all its components in a way understandable to those (like me) without much neuroscience background.

**Weaknesses:**

While there are several interesting ideas, and use of columns in CCM appears to work well, I think the links to group convolutions could be drawn out and studied even more explicitly.  See my comments below on Sec 4.7 for more details.

In addition, while there are several relevant comparison systems --- especially ResNeXt, which uses group convs, there are no comparisons to many more recent CNNs that may also be relevant.  Searching on paperswithcode.com for CNNs on ImageNet yields many from the past couple years.  (Note not all of these are necessarily relevant comparisons --- but some likely are:  the current leaderboard #1 is RevCol which also uses the term "column" extensively, though I think in a different way, while MogaNet uses similar param counts).

**Questions:**

Sec 4.7:

  * "The idea of IR in ResNeXt is more similar to us however, the major difference is that
ResNeXt has multiple blocks per stage, and each performs replication":  this could be spelled out more, as this sentence leaves the similarities vague.  ResNeXt uses group convs, but I don't think IR is mentioned in that paper --- however, IR followed by group conv is the same as a single regular convolution ($group_conv(IR(x)) = conv(x)$), so if resnext performs $group_conv(f(conv(x)) = group_conv(f(group_conv(IR(x)))$ where $f$ is normalization+activation, then indeed there is a close link, and stacking more group convs is a major difference.

  * discussion comparing to group convolutions:  This is a good discussion, particularly the second point.  However, I think it could be expanded further, and perhaps referred to even more throughout the paper.  If a "CCM" is implemented by group conv + residual, the fact that there are stacked group convs without between-group recombination could be brought more to the forefront and studied explicitly.  What happens when varying $l$ to be smaller, so that there is a 1x1 recombination more frequently, even for $l=1$ (and more blocks); or alternatively, introducing 1x1 recombinations after *every* group conv?  Is this model then the same as ResNeXt?  If so, then this is a critical difference.  Overall it would be good to nail this down more, going incrementally from ResNext-like operations (group conv followed by 1x1 recombination/mapping) to CCM.

  * "no blocks, only stages":  It's unclear in this section what constitutes the notion of a "block" and why comnet does have it --- since it does have residual connections between each of the $l$ conv applications (which are analogous to "blocks").  What is meant by "no block" here and why is it advantageous?  Does it have to do with how often there is cross-column combination?


Smaller comments on other sections:

* related work:  An older related paper, is "network-in-network" (https://arxiv.org/pdf/1312.4400.pdf), which interprets 1x1 convs as applying a small MLP within each conv window, extending plain convs beyond linear combination, in the same vein as this work interprets stacked group convs as columnar operations.

* sec 2, 4.6:  "exponential param growth" --- what does this refer to and how exactly is it exponential?  increasing depth is linear param growth, while increasing width is quadratic.


* Tables:  A0, B1, C1, etc:  I didn't see the variants described in the text, only listed in an appendix table.

* Table 2:  is this showing the right numbers?  none of the comnet models have a difference in params or flops for plain vs SR, but still a speed increase



* fig1: abstract mentions parnet next to ref to fig1, but parnet is not in this figure.

* fig 2d:  "Conv, b=M":  the parameter "b" is not described anywhere; is the number of groups in a group conv?

* eq 1:  $T_{ccm} = CCM(.)$:  I don't see anywhere this is used.  Also, would be good to provide a definition of CCM in this set of equations, expressed in terms of convs or group convs.

---

> ### Author Response · Authors · 2023-11-14
> **Answeres to the questions**
>
> We thank the reviewer for applauding the contributions and highlighting the paper's strengths. Below we try our best to resolve your questions.
>
> **Discussion on RevCOL and MogaNet**
>
> Thank you for bringing these papers to our knowledge. Definitely, these papers work in the area of CNN. However, they are different from our motivation that we intend to develop a macro ConvNet design which is *Pure CNN* i.e. no attention mechanisms or distributed information paths causing high branching. *RevCOL*, it is fundamentally different, where a column refers to a deep network which makes the final prediction. Multiples of these deep networks are used, which receive the same input and produce independent final predictions and any two columns have different weights. On the other hand,  in CoMNet, multiple columns do not produce the final output. There is only one output end of CoMNet in contrast to the RevCOL. Then If we look closely at *MogaNet*, it has several branches due to different-sized kernels and a very complex information path, resulting in lower latency. Although it happens to be the reason for its competitive results, we believe that CoMNet is a much purer form of CNN which can also be improved similarly to MogaNet by inheriting the fundamentals of CoMNet design e.g. different kernel-sized columns.
>
>
> #Questions
>
> **1. "The idea of IR in ResNeXt is more similar to us however, the major difference is that ResNeXt has multiple blocks per stage, and each performs replication": this could be spelled out more. ResNeXt uses group convs, but I don't think IR is mentioned in that paper --- however, IR followed by group conv is the same as a single regular convolution (), so if resnext performs where is normalization+activation, then indeed there is a close link, and stacking more group convs is a major difference.**
>
> Thank you for the observation. Indeed, there is a close link, but the CoMNet structure is different, as pointed out by you, i.e. *stacking more group convs is a major difference.* Since we try to follow biological observations, we have included IR to be consistent with the theory. IR followed by a group convolution is indeed a plain convolution; however, this is true only if there is one convolution. This does not apply to the entire stack in the CoMNet stage. So in CoMNet, the first 3x3 convolution after IR can be replaced by a plain convolution, but the other convolutions in the column stack have to be group conv. Another difference is that in ResNext, 1x1 convolution is used for this purpose, however, in ConvNet, it is a 3x3 conv.
>
>
> **discussion comparing to group convolutions: This is a good discussion, particularly the second point. However, I think it could be expanded further, If a "CCM" is implemented by group conv + residual, the fact that there are stacked group convs without between-group recombination could be brought more to the forefront and studied explicitly. What happens when varying to be smaller, so that there is a 1x1 recombination more frequently, even for (and more blocks); or alternatively, introducing 1x1 recombinations after every group conv? Is this model then the same as ResNeXt? If so, then this is a critical difference. Overall it would be good to nail this down more, going incrementally from ResNext-like operations (group conv followed by 1x1 recombination/mapping) to CCM.**
>
> Indeed l=1 and introducing 1x1 recombinations after every group conv makes a CoMNet bottleneck same as ResNeXt. This is what differentiates CoMNet from ResNeXt. There is a stack of 3x3 and only 8 1x1 layers are used in a CoMNet variant (CoMNet-C2) that is equivalent to ResNeXt-50 with 32 1x1 layers. To verify what happens after frequent 1x1 combinations, we experimented as shown in *Sec.C, Ablation study, Varying-l* of the paper.
>
> Based on your comments, we further elaborate the text which better shows differences with ResNeXt.
>
>
> **no blocks, only stages": It's unclear in this section what constitutes the notion of a "block" and why comnet does have it --- since it does have residual connections between each of the conv applications (which are analogous to "blocks"). What is meant by "no block" here and why is it advantageous? Does it have to do with how often there is cross-column combination?**
>
> Traditionally, a ConvNet block is referred to as a bottleneck which consists of a stack or a combination of different convolutions. Indeed CoMNet have a residual connection between the immediate 3x3 convs, however, it is different from the traditional designs that skip connection is formed across a stack of more than one convolution. For this reason, the earlier VGG also had only stages instead of blocks or bottlenecks. In the same fashion, in Comnet, there is the stage which consists of a stack of convolution, thus in CoMNet, blocks or stages can be used interchangeably.
>
> **Based on your final comments and recommendations, we will revise the text with the above explanations  until 20th**

---

> ### Author Response · Authors · 2023-11-14
> **Response to Smaller comments on other sections:**
>
> **Related work: An older related paper, is "network-in-network" (https://arxiv.org/pdf/1312.4400.pdf), which interprets 1x1 convs as applying a small MLP within each conv window, extending plain convs beyond linear combination, in the same vein as this work interprets stacked group convs as columnar operations.**
>
> We include this paper in the references.
>
> **sec 2, 4.6: "exponential param growth" --- what does this refer to and how exactly is it exponential? increasing depth is linear param growth, while increasing width is quadratic.**
> By this statement, we mean that as we go deep down, the deeper layers have a large number of channels since the width is increased. As if increase the number of repetitions of deeper blocks, together, the increased depth and the increased width causes drastical changes in the parameters. For example, ResNet-50 has roughly 14M parameters in only the final 3 blocks of stage-5 whereas the remaining 10M parameters out of 25M total parameters come from the remaining 13 blocks of the network.
>
> We'll add this explanation into the text.
>
> **Tables: A0, B1, C1, etc: I didn't see the variants described in the text, only listed in an appendix table.**
>
> This aligns with the contribution 6 in Section-1 that we not only present CoMNet study, but also its minimal design space so that it can be used by the community.
>
> **Table 2: is this showing the right numbers? none of the comnet models have a difference in params or flops for plain vs SR, but still a speed increase**
> Yes. They are correct, and in fact this is an interesting fact about the CoMNet in contrast to the RepVGG. That RepVGG have different paths and different kernels (shortcut, 3x3 and 1x1 in parallel) at the train time, which are fused by a single convolution at the test time via SR. Whereas in CoMNet, we only have one 3x3 Convolution and identity, therefore we do not encounter any parameter change during SR. The speed improvement comes from the fusion of identity or shortcut into the 3x3 Conv via SR which shows how branching can affect speed.
>
> **fig1: abstract mentions parnet next to ref to fig1, but parnet is not in this figure**
> We will incorporate the readings of ParNet into Fig.1
>
> **Fig  2d: "Conv, b=M": the parameter "b" is not described anywhere; is the number of groups in a group conv?**
> Yes. It is a typo. 'b' should be 'g'. Thank you for pointing this out.
>
> **eq 1: I don't see anywhere this is used. Also, would be good to provide a definition of CCM in this set of equations, expressed in terms of convs or group convs.**
> Applied your comment.
>
> **Based on your final comments and recommendations, we will revise the text with the above explanations until 20th**

---

> ### Author Response · Authors · 2023-11-14
> **Please mention any further comments and changes to be  made.**
>
> Dear reviewer,
>
> based on assessment of our responses, kindly mention any additional changes you would want to see in the final text.
>
> Thank you

---

> ### Comment · Reviewer_mQDh · 2023-11-21
> **responses**
>
> Thanks for your responses.
>
> My largest concerns now are the points brought up by reviewer Y3Qk.  I'm not very familiar with the background work in biology, so will defer to them for this aspect.  While it looks to me there is now a decent mutual understanding of the changes required, it seems they'd require a significant rewrite and may require a resubmit rather than minor revision.
>
> As for my initial review, most of the comments in my initial review have been addressed, except for the following:
>
> * ablation on $l$:  I don't think this is explored much in the appendix table A2:  row 9 only increases $l$, whereas the more interesting comparison would be to decrease $l$ down to 1, so it coincides with ResNext, then observe the behavior as it is increased from 1 to larger numbers.
>
> * small detail of the use of "exponential" in the phrase "exponential param growth":  It is still not exponential, a less precise term such as "large" param growth would be better here.

---

### Official Review · Reviewer_Kwm3 · 2023-10-30

**Soundness:** 3 good
**Presentation:** 3 good
**Contribution:** 2 fair
**Rating:** 5
**Confidence:** 5

**Summary:**

The paper introduces a ConvNet architecture called CoMNet, which is inspired by the structural organization of cortical modules in the biological visual cortex. CoMNet is designed to offer efficiency in multiple dimensions such as network depth, parameters, FLOPs, latency, branching, and memory budget simultaneously. The authors propose a Multi-Dimensional Efficiency (MDE) evaluation protocol to test the model. Through comprehensive evaluations, CoMNet is shown to outperform several representative ConvNet designs including ResNet, ResNeXt, RegNet, RepVGG, and ParNet in the MDE setting.

**Strengths:**

1. The idea is inspired from the visual cortex. The authors explore the Columnar Structure, Shared Input or Input Replication, Limited Synaptic Connections, Massive Parallelization and Lateral Connection Inhibition which are refer to the biological visual cortex. Comprehensively , the authors club these valuable cortex properties into one architecture through a systematic study.
2. The authors build their template ConvNet step-by-step. They first combine the fundamental design attributes of the cortex design and then translate those attributes into its neural equivalent. Futher, they develop its CNN equivalent and develop the fundamental computational
 CoMNet-unit. This construct process is sequentially and well founded.
3. In order to improve the model's efficiency, the authors focus on $5$ crucial dimensions include latency, depth, branching, FLOPs and parameters. Through careful design, the CoMNet is both with powerful representation ability and hardware efficiency.

**Weaknesses:**

1. Just as the authors mentioned, most of the ideas inspired from the visual cortex have been used in ConvNets, such as weight sharing, shortcut, groups convolution, etc. For each design point of this article, it is not novel. Generally, the novelty of this paper is weak.
2. Inspired from the biological function, there are many frameworks to simulate the structure. The authors and previous works provide different solutions, and the designed modules are based on some hypothesis which are not proved.
3. The experiments is insufficient.

**Questions:**

1. The experiments only compare with partial works, all works should be compared in different level, such as efficientnet is only exist in R1 and Parnet only in R4.
2. One more ablation study, the function of the number of lateral connections or the number of stages.
3. The results in Table 1 is doubtful, why R0, R1, R2 and R3 have the same latency?  From my experience, it is impossible.
4. In Table 2, what is the method that combines CoMNet with Structural Reparameterization?

---

> ### Author Response · Authors · 2023-11-14
> **Responses**
>
> Dear Reviewer,
>
> We thank you for acknowledging the strengths and competitiveness of CoMNet. Below, we address your concerns.
>
> # Weakness
>
> **1. Just as the authors mentioned, most of the ideas inspired from the visual cortex have been used in ConvNets, such as weight sharing, shortcut, groups convolution, etc. For each design point of this article, it is not novel. Generally, the novelty of this paper is weak.**
>
> We appreciate your concern. However, it must be noticed that despite weight sharing, shortcuts, and group convolution are prominent in CNNs, many essential cortex properties are still missing from the CNN designs. Our paper is the first to club almost all of them in one architecture and present multi-dimensional efficiency in CNNs. The major novelty from the previous design is using 3x3 convolutions by majority and following the columnar approach. Also, avoiding the need for bottleneck designs which leads to only stage and no blocks concept in CoMNet. Together, these improvements over the existing models are major novelty which helps comnet achieve multidimensional efficiency, including transformers.
>
> **2. Inspired from the biological function, there are many frameworks to simulate the structure. The authors and previous works provide different solutions, and the designed modules are based on some hypothesis which are not proved.**
>
> We agree with your comment and this also applies to well-established CNN models and Transformers. At least at some point, these models draw inspiration from biology but these inspirations are not well connected. However in this paper, we concretely revisit the old works in the area of visual cortex, and try to align the modern ConvNet design process with these ideas. Despite neuroscience has advanced so much, however, a concrete pin-pointed architecture of cortex is still not available in literature. This forces the network design researchers to draw their own conclusions and designs which try to mimic the biological cortex. In the same sense, we do not claim or bring anything new about the cortex, instead borrow simple design ideas of cortex, and try to develop a simple CNN architecture which works well on multiple fronts. We verify the same via a number of experiments.
>
> **The experiments only compare with partial works, all works should be compared in different level, such as efficientnet is only exist in R1 and Parnet only in R4.**
> The reason behind this choice is the resolution. For example, only efficientnet-b0 model is trained and tested at 224x224 resolution which is a standard. Other variants of efficientnet are trained at higher resolution via compound scaling which improves the accuracy due to the availability of more information. On the other hand, we presented our model only at 224x224 which is a very general trend, and we do not study compound scaling on comnet as it would be a separate line of work.
>
> While in terms of ParNet, these models are big. In order to compare with them, we kept the accuracy as a constant and chose suitable comnet variants which can match in accuracy with ParNet. It can be seen that CoMNet is extremely fast than the recent ParNet series of work.
>
> **One more ablation study, the function of the number of lateral connections or the number of stages.**
>
> It is a common tradition to reduce the resolution of ImageNet images at 224x224 by a stride of 32 which forms five stages. Therefore we also follow the same tradition of five stages among which first is stem and the remaining four are CoMNet units.
> While the effect of lateral connections can be observed in the Table-A2 Appendix and comparing R0 with R3. R3 has doubled the number of lateral connections which increases the parameter count and hence accuracy.
>
>
>
> **The results in Table 1 is doubtful, why R0, R1, R2 and R3 have the same latency? From my experience, it is impossible.**
> The results on R0-R3 are absolutely certain and correct. The confusion arises from the fact that it is a general trend to increase the depth to achieve higher accuracy. However in CoMNet, we change the number of modules and slightly the depth. By increasing the number of modules, we are essentially modifying the width. Moreover, since depth is roughly the same, the whole computations of CoMNet remains packed into layers thus there are more parallel computations with high computation density in each layer of CoMNet and less serial operations due to smaller depth. This causes similar latency in different CoMNet variants despite having different capacities.**
>
> **In Table 2, what is the method that combines CoMNet with Structural Reparameterization?**
> We simply fuse the identity connection in CoMNet unit similar to RepVGG and now run the inference. Fusing the identity connection with 3x3 conv is essentially the SR operation.
>
> **We once again thank you for your concerns. We will apply your comments and suggestions in the revised text until the 20th after hearing back from you on the responses.**

---

### Official Review · Reviewer_nTg9 · 2023-10-31

**Soundness:** 3 good
**Presentation:** 3 good
**Contribution:** 2 fair
**Rating:** 6
**Confidence:** 2

**Summary:**

**I am not very familiar with this field, so my evaluation may have limited reference value.**

This paper presents CoMNet, a ConvNet architecture inspired by the structural organization of the biological visual cortex. CoMNet is described as a simplified yet powerful design that offers efficiency across multiple dimensions such as network depth, parameters, FLOPs, latency, branching, and memory budget.

**Strengths:**

1. The paper is well-written and presents its ideas clearly.
2. The proposed model achieves higher performance with fewer computation cost and latency.

**Weaknesses:**

1. The article only compares earlier works, with the most recent being from 2021. I believe that comparing CoMNet with some more recent methods could enhance the quality of the paper.
2. The comparison with Vision Transformer (ViT) could be included in the main body of the paper instead of the appendix. Additionally, some of the latest backbones could be used for ViT.
3. The paper could also benefit from conducting experiments in more areas, such as segmentation and detection. Such experiments might further enhance the quality of the paper.

**Questions:**

Please see weaknesses.

---

> ### Author Response · Authors · 2023-11-14
> **Responses**
>
> Dear Reviewer,
>
> We thank you for acknowledging the advantages of our model. Below we address your concerns.
>
> **1. The article only compares earlier works, with the most recent being from 2021. I believe that comparing CoMNet with some more recent methods could enhance the quality of the paper.**
> The reason for this being that our CoMNet is a pure CNN design i.e. free of advance information paths or attention mechanism. We compared the most recent works uptill 2022 (ParNet) in pure CNN realm.
>
> **2. The comparison with Vision Transformer (ViT) could be included in the main body of the paper instead of the appendix. Additionally, some of the latest backbones could be used for ViT.**
> We will move the results to the main text.
>
> **3. The paper could also benefit from conducting experiments in more areas, such as segmentation and detection. Such experiments might further enhance the quality of the paper.**
> We agree with you. Since this is a new design with multiple concepts, we tried to arrange our experimental setup such that it can be understood easily by the readers instead of incorporating many areas into one paper.
>
> **We once again thank you for the comments. Please provide additional comments to our responses if any. We ll apply your comments to the final text until 20th based on your comments to our responses**

---

> > ### Comment · Reviewer_nTg9 · 2023-11-23
> >
> > Dear Authors,
> >
> > Thanks for your responses. I would like to help you further improve the quality of the paper but I am really not familiar with this area. So I would like the maintain my original positive score.

---

### Official Review · Reviewer_Y3Qk · 2023-11-01

**Soundness:** 1 poor
**Presentation:** 1 poor
**Contribution:** 3 good
**Rating:** 3
**Confidence:** 3

**Summary:**

The paper presents a new convolutional network block inspired by some properties of biological cortical columns. The architecture is found to perform well in image recognition in multiple dimensions of interest including accuracy, FLOPS, parameter count, etc.

**Strengths:**

The paper simultaneously analyses several metrics that are diverse and reasonable.

The method is compared with strong baselines including widely used convolutional networks and a couple of newer networks and shows competitive outcomes.

**Weaknesses:**

The architecture is motivated by certain biological details of the cortex, but these details are largely mischaracterized. Section 3, which reviews the biological visual cortex, has many errors. For example, ocular dominance columns are referred to as “the modules in shallower layers” whereas they coexist with orientation columns. Mountcastle et al. (1997) is given as a reference for the statement, “A cortical column contains only a few neurons (70−100)” but Mountcastle refers to this unit of organization as a minicolumn, and says that a column has many minicolumns. It is claimed that the small number of neurons in a 70-100 unit “column” limits neuron connectivity, but cortical cells have orders of magnitude more connections than that. The next paragraph claims that receptive fields are small regardless of depth of a neuron within the cortex, but deeper neurons (e.g. in medial superior temporal or inferotemporal cortex) have receptive fields tens of degrees wide. There are a number of other examples as well.

There is a further misunderstanding in the statement, “Cortical modules can not communicate with each other, except at their output (Tanaka, 1996) via pyramidal neurons.” This is a reasonable statement in the context of a feedforward model, but the authors miss the fact that a strong majority of cortical cells are pyramidal. Instead they associate pyramidal cells with a 1x1 convolution at the output of a multi-layer module in which the majority of model neurons aren’t pyramidal.

There are two problems with this general lack of grounding of the architecture in neuroscience. First, as it’s written the paper is misleading with respect to neuroscience. Second, there is no remaining motivation or justification for the architecture.

In summary, the architecture works well, but it is introduced in a confusing and misleading way. In my opinion, references to biology should be purged, the architecture should be explained in terms of the key innovations relative to prior networks, and these contributions should be analyzed in a revised ablation study.

**Questions:**

The ablation study in the appendix shows that skip connections and width improve performance, but otherwise I don’t have a clear sense of why this architecture works well. If I understand correctly the main novelty is that each stage has multiple parallel streams with the same structure. Compared with Inception blocks, they have greater depth and more uniformity. Is that the main innovation in the architecture? Does that organization account for the network’s good performance relative to RepVGG?

---

> ### Author Response · Authors · 2023-11-14
> **Responses**
>
> Dear Reviewer
>
> We thank you for your acknowledging the strong experimental evaluations and the usefulness of the architecture. Below we address your concerns.
>
> **The architecture is motivated by certain biological details of the cortex, but these details are largely mischaracterized. Section 3, which reviews the biological visual cortex, has many errors. For example, ocular dominance columns are referred to as “the modules in shallower layers” whereas they coexist with orientation columns.**
>
> We believe that there is slight misunderstanding in reading the text. By occular dominance columns we indeed mean orientation columns. Occular dominance columns are also known as orientation columns. We also have mentioned in the main text that they respond to orientation of simple stimuli such as edges.
>
> **Mountcastle et al. (1997) is given as a reference for the statement, “A cortical column contains only a few neurons (70−100)” but Mountcastle refers to this unit of organization as a minicolumn, and says that a column has many minicolumns.**
> It is a typo in our paper. Definitely a cortical module is a combination of minicolumns. That is why in our paper we have a concept of ACM and CCM. stacked ACMs mimic minicolumns having only a fewer number of neurons.
>
> **It is claimed that the small number of neurons in a 70-100 unit “column” limits neuron connectivity, but cortical cells have orders of magnitude more connections than that.**
> We agree with you but it is only true for pyramidal neurons. If we have a look at the Figure-16, Tanaka et al. mentions the same that each column has smaller number of neurons which are fused at the end via pyramidal neurons
>
> **The next paragraph claims that receptive fields are small regardless of depth of a neuron within the cortex, but deeper neurons (e.g. in medial superior temporal or inferotemporal cortex) have receptive fields tens of degrees wide. There are a number of other examples as well.**
> To our knowledge, we have not made such a claim in Sec 3.
>
> **Cortical modules can not communicate with each other, except at their output (Tanaka, 1996) via pyramidal neurons.**
> We agree with you that pyramidal neurons are more often present in the cortex but cortical column outputs are fused via pyramidal cells as mentioned on Page-15 Tanaka et al. Unfortunately, the behaviour is still difficult to mimic because it causes a very large number of branching and also pyramidal cells project their input to different levels. So in all this behavior is still far from realization from a hardware efficiency standpoint.
>
> **There are two problems with this general lack of grounding of the architecture in neuroscience. In summary, the architecture works well, but it is introduced in a confusing and misleading way. In my opinion, references to biology should be purged, the architecture should be explained in terms of the key innovations relative to prior networks, and these contributions should be analyzed in a revised ablation study.**
>
> We sincerely partly disagree with you because almost all of the models are, to a point, inspired by the visual cortex e.g. LeNet. However, due to the lack of a pin-pointed cortex structure, it is difficult to build an architecture which perfectly mimics the cortex structure. Same is with CoMnet but the difference is that we tried to bring more ideas into a single convnet design from the cortex.
>
> We are open to rewriting the contents of the paper, especially the biology section, but unfortunately, it can not be removed entirely since the paper's architecture is completely based on those theoretical and experimental evidences provided by the biological references.
>
> **The ablation study in the appendix shows that skip connections and width improve performance, but otherwise I don’t have a clear sense of why this architecture works well. If I understand correctly the main novelty is that each stage has multiple parallel streams with the same structure. Compared with Inception blocks, they have greater depth and more uniformity. Is that the main innovation in the architecture? Does that organization account for the network’s good performance relative to RepVGG?**
>
> There are multiple novelties of the architecture which makes it possible for CoMNet to achieve efficiency on multiple fronts. Yes, one of the novelties is deeper stages and uniformity with precisely controllable depth. The columns are decoupled from each other until last fusion 1x1 convolution layer. This allows the network to learn valuable representation without getting overfitted due to small number of synaptic connections. This is in line with the motivation of depthwise separable convolutions.
>
> Smaller connections, smaller depth due to the elimination of 1x1 convs let CoMNet achieving its MDE goal.
>
> **We again thank you for your comments. Please provide comments to our responses and we apply all of them to the final text until 20th**

---

> > ### Comment · Reviewer_Y3Qk · 2023-11-16
> >
> > Orientation columns and ocular dominance columns are not the same. Orientation columns refer to groups of neurons with similar orientation selectivity via Gabor-like receptive fields, whereas ocular dominance columns relate to which eye drives a group of neurons more strongly. Again, most of the neuroscience in the paper (like the claim that "Occular dominance columns are also known as orientation columns"), is inaccurate.
> >
> > You are right that convolutional networks generally have a number of parallels with visual cortex. However, I can't discern additional parallels in your model. Your model does bear a notable resemblance to ResNeXt, so it would make sense to introduce it in that context instead. Similarly, the ResNeXt paper explained itself in terms of ResNets rather than neuroscience.
> >
> > You have strong results, but I want to be as clear as I can that the neuroscience material is thoroughly unacceptable and should be thoroughly removed.

---

> > > ### Author Response · Authors · 2023-11-16
> > > **Only biological references to be purged along with section 3 or the notion of ACM and CCM, long range projections etc?**
> > >
> > > Dear Reviewer,
> > >
> > > We are highly encouraged due to your positive feedback. However we want some clarifications regarding paper edit. We have below questions regarding the presentation:
> > > 1. Since in the introduction, we specify the notion of artificial cortical modules and columnar organization and our biological inspirations. Shall be purge them or keep them as it is since they do not say anything about neuroscience?
> > > 2. Sec 3 will be removed completely. However, is it OK to use the terms of ACM and CCM in the remaining text?
> > > 3. Do You mean to purge only section-3 and the associated biological references throughout the paper?
> > > 4. So far, we have understood that i) remove section 3 and the references while the rest of the paper i.e. inspiration from biology and the terms like cortical modules, columnar organization, and pyramidal neurons, can be kept intact but without biological references. Is that what you suggest?
> > >
> > > We kindly request you to respond to each of the queries as soon as possible so we can prepare a revised edit based on your and other reviewers' suggestions.
> > >
> > > Thank you

---

> > > > ### Comment · Reviewer_Y3Qk · 2023-11-16
> > > >
> > > > I think referring to any ideas from neuroscience without a suitable foundation will detract from the paper and confuse readers. The terms "cortical" and "pyramidal" are hard to interpret any other way, unfortunately. In place of "pyramidal neurons", "projection neurons" is a more general idea from neuroscience that I think expresses what you want without being problematic. A projection neuron is one that has an outgoing connection outside a local circuit, for example to a different nucleus or cortical area. Apart from that, I think it would be clearer to use the same terminology as ResNeXt wherever possible. If you want to emphasize the longer series' of convolutions, you could continue to call these "columns" since they appear as architectural columns independent of any similarity to cortex. I hope that is helpful.

---

> > > > > ### Author Response · Authors · 2023-11-16
> > > > > **Thank you so much**
> > > > >
> > > > > We thank you so much for your constructive feedback. We'll apply all of your suggestions into the final text.

---

> > > > > > ### Comment · Reviewer_Y3Qk · 2023-12-04
> > > > > > **Serious issues unaddressed**
> > > > > >
> > > > > > The current version of the paper still contains many incorrect and misleading claims about neuroscience and lacks clear motivation for the architecture relative to similar networks such as ResNeXt. A substantial rewrite is needed, so perhaps the authors want to take more time to revise the paper and submit it somewhere else. In any case, it can't be accepted in its current form. The other reviewers did not express the same concerns as me about the neuroscience content, but I suspect that is due to their different experiences that may include less neuroscience. This reinforces my view that the paper (as currently written) is actively misleading as well as being incorrect. It is unfortunate because the results seems strong.

---

### Meta-Review · Area_Chair_3meg · 2023-12-05

**Metareview:**

This paper describes a convolutional network block supposedly inspired by neuroscience. The architecture performs well for image categorization as measured according to multiple dimensions of interest, including accuracy, FLOPS, parameter count, etc. However, the reviewers pointed to a number of weaknesses. One is that the contribution of this paper is limited to what appears to be relatively small and incremental changes to existing CNN architectures. These contributions were judged to be below the bar for acceptance at ICLR. There was also strong pushback from some reviewers regarding the neuroscience claims, and the AC agrees with these criticisms. Because these neuroscience claims provide the main motivation for the main design choices, the paper was judged to be below the threshold for acceptance. The AC thus recommends the paper be rejected.

**Justification For Why Not Higher Score:**

This work has several weaknesses. The design choices that are motivated by neuroscience claims appear to be flawed. More problematic, the overall innovation and intellectual contributions appear well below the threshold for ICLR.

**Justification For Why Not Lower Score:**

NA

---

### Decision · Program_Chairs · 2024-01-16

Reject